# Study on microwave ablation temperature prediction model based on grayscale ultrasound texture and machine learning

Yan Xiong[1], Yi Zheng[2], Wei Long[1], Yuxin Wang[2], Qin Wang[1], Yi You[1], Yuheng Zhou[1], Jiang Zhong[3], Yunxi Ge[3], Youchen Li[3], Yan Huang[3]*, Zhiyong Zhou[2,4]*

1 Nanjing University of Chinese Medicine, Nanjing, China, 2 Suzhou Institute of Biomedical Engineering and Technology, Chinese Academy of Science, Suzhou, China, 3 Department of Ultrasound, Nanjing Hospital of Chinese Medicine Affiliated to Nanjing University of Chinese Medicine, Nanjing, China, 4 School of Biomedical Engineering (Suzhou), Division of Life Sciences and Medicine, University of Science and Technology of China, Suzhou, China

☯ These authors contributed equally to this work.
* jacob6666@163.com (YH); zhouzy@sibet.ac.cn (ZZ)

## Abstract

### Background

Temperature prediction is crucial in the clinical ablation treatment of liver cancer, as it can be used to estimate the coagulation zone of microwave ablation.

### Methods

Experiments were conducted on 83 fresh ex vivo porcine liver tissues at two ablation powers of 15 W and 20 W. Ultrasound grayscale images and temperature data from multiple sampling points were collected. The machine learning method of random forests was used to train the selected texture features, obtaining temperature prediction models for sampling points and the entire ultrasound imaging area. The accuracy of the algorithm was assessed by measuring the area of the hyperechoic area in the porcine liver tissue cross-section and ultrasound grayscale images.

### Results

The model exhibited a high degree of accuracy in temperature prediction and the identification of coagulation zone. Within the test sets for the 15 W and 20 W power groups, the average absolute error for temperature prediction was 1.14°C and 4.73°C, respectively. Notably, the model's accuracy in measuring the area of coagulation was higher than that of traditional ultrasonic grey-scale imaging, with error ratios of 0.402 and 0.182 for the respective power groups. Additionally, the model can filter out texture features with a high correlation to temperature, providing a certain degree of interpretability.

**Data Availability Statement:** All relevant data are in the paper and its Supporting Information files.

**Funding:** This work was supported partly by National Natural Science Foundation of China (62271480), Youth Innovation Promotion Association CAS (2021324), Jiangsu Key Technology Research Development Program (BE2021612), Science and Technology Development Project of Suzhou (SYG202321), Postgraduate Research and Practice Innovation Program of Jiangsu Province (KYCX23_2093). The funders had no role in study design, data collection and analysis, decision to publish, or preparation of the manuscript.

## Conclusion

The temperature prediction model proposed in this study can be applied to temperature monitoring and coagulation zone range assessment in microwave ablation.

## 1. Introduction

As the sixth most common cancer worldwide and the third leading cause of cancer death, primary liver cancer accounted for over 900,000 new cases globally in 2020 alone [1]. Despite a recent decline in the incidence of liver cancer, the poor prognosis and high mortality rate continue to pose a significant medical burden [2], with hepatocellular carcinoma (HCC) accounting for 90% of primary liver malignancies [3] and a five-year overall survival (OS) rate of merely 20% for HCC [4]. Currently, surgical resection, liver transplantation, and ablation therapy are the three main curative treatments for early-stage HCC [4]. Microwave ablation (MWA), as an emerging ablation modality, induces coagulative necrosis and cell death by heating tissue through electromagnetic wave-induced vibration of water molecules [5]. Compared to traditional radiofrequency ablation, MWA's active heating method is unaffected by impedance, achieving shorter ablation times, higher temperatures, and larger coagulation zone, with less impact due to the heat sink effect [5–7]. Although MWA has been widely used in liver cancer treatment, difficulty in controlling the energy in the coagulation zone remains a major factor limiting the prognosis of ablation surgery. Insufficient temperature can lead to tumor residue and increased postoperative recurrence risk, while excessive temperature may result in the burning of adjacent normal tissues due to MWA's high thermal efficiency, leading to a series of complications such as bleeding, bile duct damage, and gastrointestinal perforation [8]. The temperature distribution within the tissue determines the range of tissue damage during ablation, with isotherms between 60°C and 100°C outlining the approximate range for complete tumor necrosis [9], making temperature monitoring in thermal ablation an important aspect of the ablation process.

Magnetic Resonance Imaging (MRI) is considered the "gold standard" among non-invasive thermometry techniques. Thanks to the high temperature dependence of MRI-related tissue parameters, including relaxation parameters (T1, T2) and proton resonance frequency, MRI can perform quantitative temperature monitoring with high spatiotemporal resolution [10]. However, its high cost and the need for ablation methods to be compatible with MRI's strong magnetic field are major barriers to its clinical application. Computed Tomography (CT) imaging can avoid these limitations. The correlation between CT value attenuation and temperature was demonstrated as early as 1979 [11], but CT thermometry requires multiple scans, exposing patients to higher radiation [12]. Compared to CT and MRI, ultrasound imaging may have a slightly inferior spatial resolution but offers unparalleled convenience. Ultrasonic thermometry is a method for inferring tissue temperature information by analyzing the echo characteristics or texture information changes of ultrasonic waves propagating in tissues. The raw echo signal, also known as the ultrasonic radiofrequency (RF) signal, forms the basis for temperature measurement methods, which primarily include time-shift method [13–15], backscatter energy method [16–18], Nakagami parameter imaging method [19–21], among others. In contrast, temperature measurement methods based on texture information changes in images [22, 23] utilize the grayscale variations and interrelationships of pixels in ultrasonic images to estimate temperature changes [24]. Texture information changes refer to the variations in the distribution and patterns of pixel intensities in ultrasound images, which are

associated with the microstructural changes in tissue induced by thermal therapy. By quantitatively analyzing these texture features, we can estimate the temperature distribution within the tissue. Due to the convenience of grayscale image acquisition, this method exhibits broader clinical application prospects compared to other thermometry techniques. Temperature changes are reflected in texture features as variations in multiple rather than single texture parameters. To assess the importance of these parameters in temperature prediction, we introduced the random forest method from machine learning to analyze these parameters through weighted voting.

This study proposes an ultrasound temperature prediction machine learning method based on radiomics, which involves first-order statistical analysis and higher-order texture feature extraction of ultrasound images around multiple thermometry points. The random forest algorithm is used to select the ten most important texture features and establish a model correlating these features with temperature, aiming to predict the temperature within a specific ultrasound imaging area.

## 2. Experimental setup and data collection

### 2.1 Experimental equipment

Ultrasound Equipment: The Vinno G86 medical ultrasound machine, manufactured in Suzhou, China, employs the X6-6L model linear array transducer for signal transmission and reception. For grayscale signal collection, the parameters are as follows: line density set to 128, the ultrasound probe frequency of 12 MHz, and a frame rate of 15 frames per second. The system operates in fundamental imaging mode with spatial compounding deactivated.

Ablation Equipment: The Ruibo microwave ablation therapy instrument, manufactured in Nanjing, China, is model MWD-A. It operates at a frequency of 2450 MHz and has a maximum output power of 100 W. The sterile disposable microwave ablation needle is model WGP-Z03, with a diameter of 1.5mm and a length of 100mm.

Thermometer: The Keysight 34970A, used for temperature measurements, samples at a frequency of 2 Hz through channels 101 to 107 using T-type thermocouple sensor wires. To ensure accurate temperature monitoring during the ablation process, a cold circulation system with 0.9% saline solution was employed.

### 2.2 Biological materials

Fresh porcine livers were trimmed to conform to the dimensions of the mold. They were then submerged in a water bath maintained at a temperature of 37°C, with a water depth of 8 centimeters. This procedure was employed to achieve a uniform heating of the liver samples, ensuring that their temperature exceeded 36°C. The mold was utilized to secure the position of the porcine liver during the heating process, ensuring consistency in the experimental conditions.

### 2.3 Temperature measurement mold and thermocouple needles

Due to the high density of biological materials, hollow needles (length: 97mm, diameter: 1.45mm) were used as the outer shell to guide the embedded thermocouples accurately to the predetermined positions, as shown in Fig 1A. To ensure that all temperature measurement points and the ultrasound imaging plane for image collection were in the same plane during the ablation process, a mold was designed to fix the temperature measurement needles and the ablation needle, as shown in Fig 1B. The mold, with specifications of $20 \times 18 \times 10$ cm, provided puncture holes for simultaneous access of one ablation needle and seven temperature measurement needles. The relative positions of the temperature measurement points and the

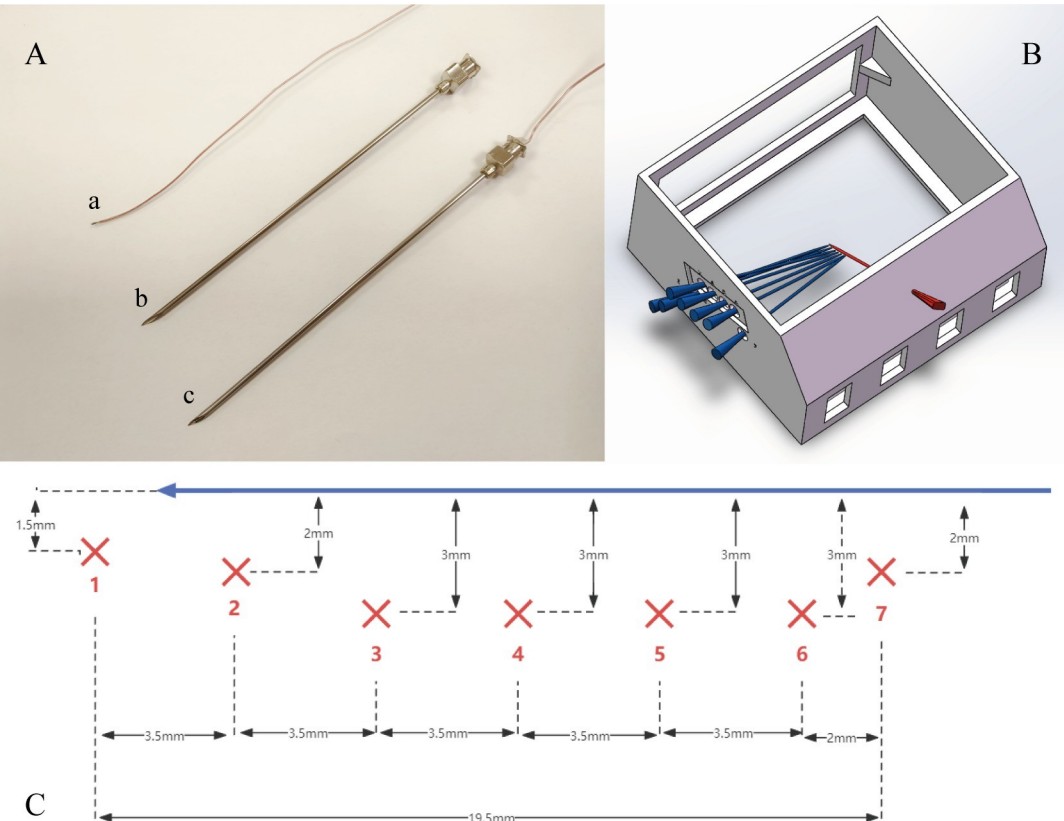

**Fig 1. Probe assembly, mold design, and positioning schematic.** (A) Temperature probe assembly details: a. Thermocouple wire—Responsible for transmitting temperature signals. b. Hollow needle tip—Serves as the protective and guiding part for the sensor wire. c. Assembled temperature probe—Demonstrates the assembly of the sensor wire and needle tip. (B) Mold design for temperature probes and ablation needle: The diagram shows the expected layout of the temperature probes and ablation needle within the mold, with the ablation needle marked in red and the temperature probes marked in blue. (C) Expected relative positions of the temperature measurement points: This details the precise layout of the temperature measurement points relative to the ablation needle. The ablation needle is indicated by blue arrows (tip to the left), the temperature measurement points are represented by red dots (indexed at the bottom), and black lines denote the expected spatial distances between the measurement points.

ultrasound scanning plane are shown in Fig 1C. Based on pre-experiments on the fusiform coagulation zone of porcine liver, the positions of the temperature measurement points were set, with all temperature measurement holes located below the ablation needle. Point 1 was located to the left of the tip of the ablation needle, and the rest were evenly distributed near the ablation needle to ensure more effective acquisition of temperature changes at different positions within the entire coagulation zone.

## 2.4 Experimental procedure

Fresh porcine livers were cut into tissue blocks with a volume of $6 \times 6 \times 4$ cm to fit the mold size and placed in a constant temperature water bath at 37°C, along with the mold, to heat the porcine liver to above 36°C. The ultrasound transducer was closely attached to the liver surface, and a region with uniform grayscale was selected. The ablation needle and temperature measurement needles were inserted along the mold. The microwave ablation instrument parameters were set (heating power P = 15 W, 20 W; heating time t = 180 s). The positions of each temperature measurement needle were marked and recorded in the ultrasound images.

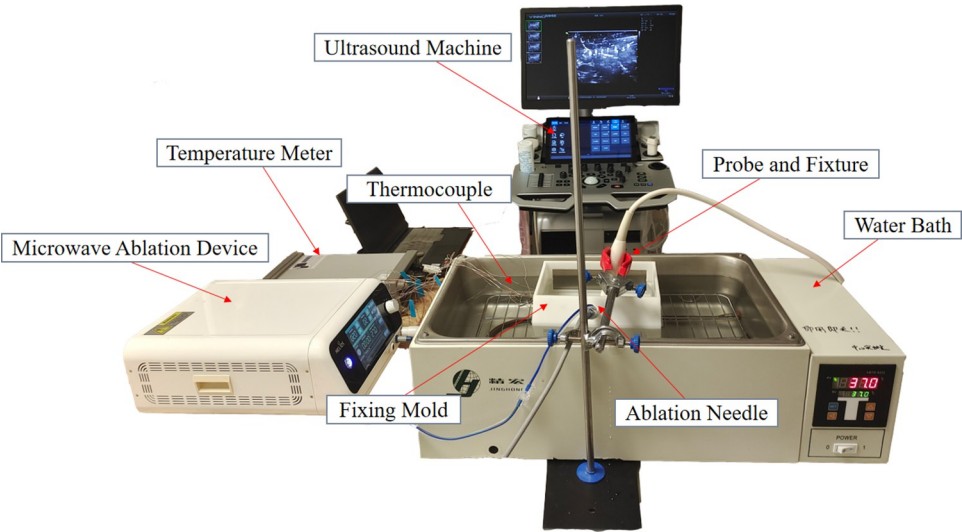

**Fig 2. Experimental setup photo.**

After the ablation, ultrasound recorded videos and multi-point temperature data were obtained, and photos of the liver tissue sections at the end of the ablation were taken. A total of 42 and 41 ex vivo porcine liver microwave ablation experiments were conducted at the two power levels, respectively. Fig 2 shows the actual experimental setup.

## 3. Dataset and methodology

Fig 3 illustrates a series of processing steps applied to the raw data. Initially, keyframes with corresponding temperature data are extracted from the ultrasound ablation videos. Based on these keyframes, the regions of interest (ROI) containing the temperature measurement points are accurately segmented. These temperature-ROI sample pairs constitute the input portion of the temperature prediction model. Subsequently, imaging biomarker features are extracted

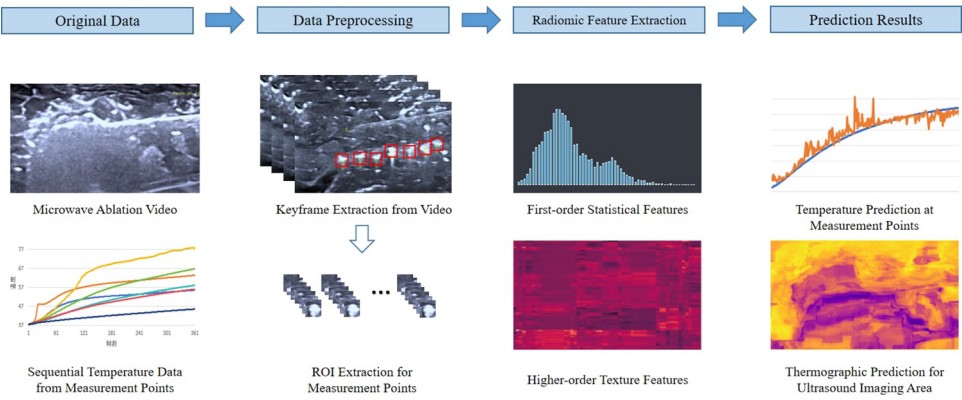

**Fig 3. Ultrasound temperature prediction method.** This figure outlines the methodological framework for extracting keyframes from ultrasound videos with corresponding temperature data. It then details the precise segmentation of regions of interest (ROIs) around temperature measurement points. The figure further illustrates the process of extracting imaging biomarker features from these ROIs, including comprehensive first-order statistical features and higher-order texture features. These features play a crucial role in capturing the details of temperature changes within the ablation zone and are subsequently used to generate a temperature estimation prediction model.

from these ROIs, encompassing first-order statistical features and higher-order texture features. These features provide rich image information, facilitating the model's capture of temperature variations. Ultimately, leveraging these features, temperature at the measurement points is predicted, and thermal maps of the ultrasound imaging region are generated based on these prediction data.

## 3.1 Preprocessing of experimental data and dataset construction

**3.1.1 Preprocessing of temperature data.** Given the multiple temperature measurement channels involved in the experiment, a varying preheating time is required for the initiation of temperature data collection by the thermometers. To synchronize data collection between the thermometers and the ultrasound machine, and to allow sufficient reaction time for the activation of the ultrasound and ablation devices, we first initiated the thermometers. Temperature data collection commenced at the 15th second after the preheating period, simultaneously with the start of ultrasound device video recording and the activation of the ablation device. The temperature data corresponding to the ablation process (from 15 seconds to 195 seconds) was retained.

**3.1.2 Preprocessing of ultrasound images.**

1. *Frame extraction from ultrasound videos*. The ultrasound videos captured during the experiment had a frame rate of 12 frames per second. To synchronize the temperature and image data, we extracted the first and sixth frames of each second from the video, corresponding to the temperature data collection frequency of twice per second.

2. *Segmentation of ROI at the temperature measurement points*. Based on the coordinates of the temperature measurement points marked in the ultrasound images before ablation, a square region of $64 \times 64$ pixels centered on the marked points was designated as ROI. Each extracted frame was segmented accordingly, as shown in Fig 4. The temperature data and ROI were matched according to the numbering recorded during the data collection process, ultimately forming temperature-ROI sample pairs.

## 3.2 Extraction of radiomic features and construction of the temperature prediction model

In the analysis utilizing the Intelligent Medical Imaging Assistant System [25], we performed extensive texture feature extraction on the regions of interest (ROIs) around the temperature measurement points using radiomics methods. These features included not only first-order statistical features (n = 19) that describe the grayscale distribution of the image, but also 2D shape-based features (n = 10) that help quantify the geometric properties of the image. Additionally, we extracted various higher-order texture features that describe the relationships between pixels, including the Gray Level Co-occurrence Matrix (n = 24), Gray Level Run Length Matrix (n = 16), Gray Level Size Zone Matrix (n = 16), Neighbouring Gray Tone Difference Matrix (n = 5), and Gray Level Dependence Matrix (n = 14). These features collectively reflect the complexity of the image texture and provide critical information for predicting the temperature distribution within the coagulation zone during microwave ablation. The radiomic features were normalized and subjected to feature selection. The selected features were then used as inputs to train the ultrasound image temperature prediction model, with temperature as the output.

In this study, feature importance was assessed using the Gini importance index [26] from the random forest algorithm [27]. The Gini importance index measures a feature's ability to

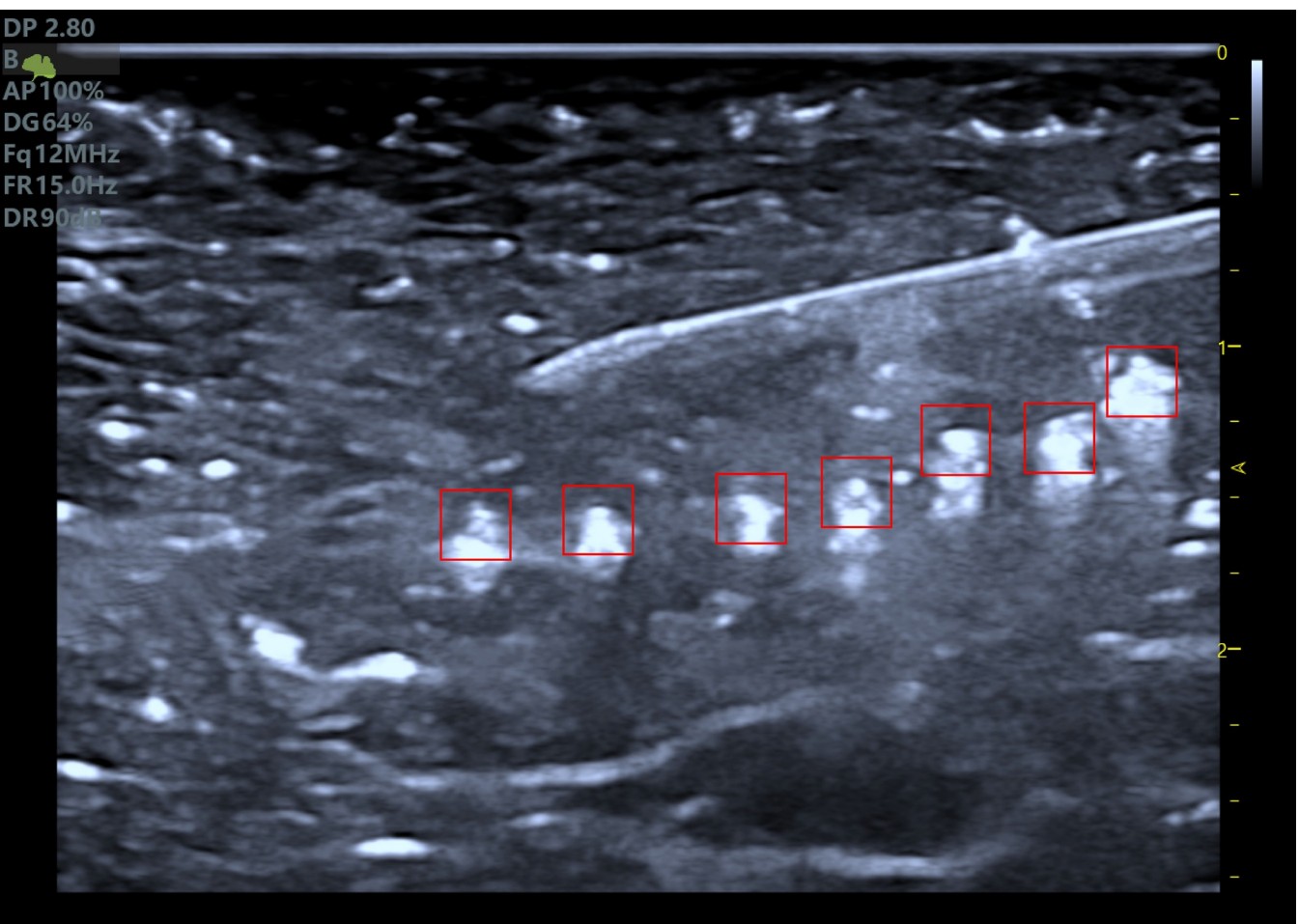

**Fig 4. Ultrasound image showing the ablation needle and temperature measurement ROIs.** This pre-ablation image captures the initial setup, with the ablation needle positioned accurately and the ROIs for temperature measurement clearly marked. Each 64 × 64 pixel square ROI is centered on a temperature measurement point, indicating the areas where temperature changes will be monitored throughout the ablation process.

reduce impurity at each split point within the random forest. A high Gini coefficient indicates that the feature provides more information during classification or regression, thus contributing more significantly to the model's predictive power. By calculating the Gini coefficient for each feature, we ranked their importance accordingly. (Fig 5) and selected the top 10 texture features in terms of importance. We believe these features are highly correlated with temperature. These features were then used to construct the temperature prediction model, which was fitted with temperature data through regression analysis. The model optimization was based on performance on the validation set to ensure optimal performance in temperature prediction.

The experiment collected and organized data from 42 cases with an ablation power of 15W and 41 cases with 20W. Other experimental parameters were consistent across all cases. The temperature data collection frequency was 2Hz, with seven temperature measurement points, and the ultrasound video frame rate was 12Hz, with a total duration of 3 minutes. Key frames were extracted from the videos based on the fixed preheating time and temperature data collection frequency, resulting in 360 frames of ultrasound image data. The temperature at the seven measurement points for each frame was obtained, forming a total of 2520 temperature-

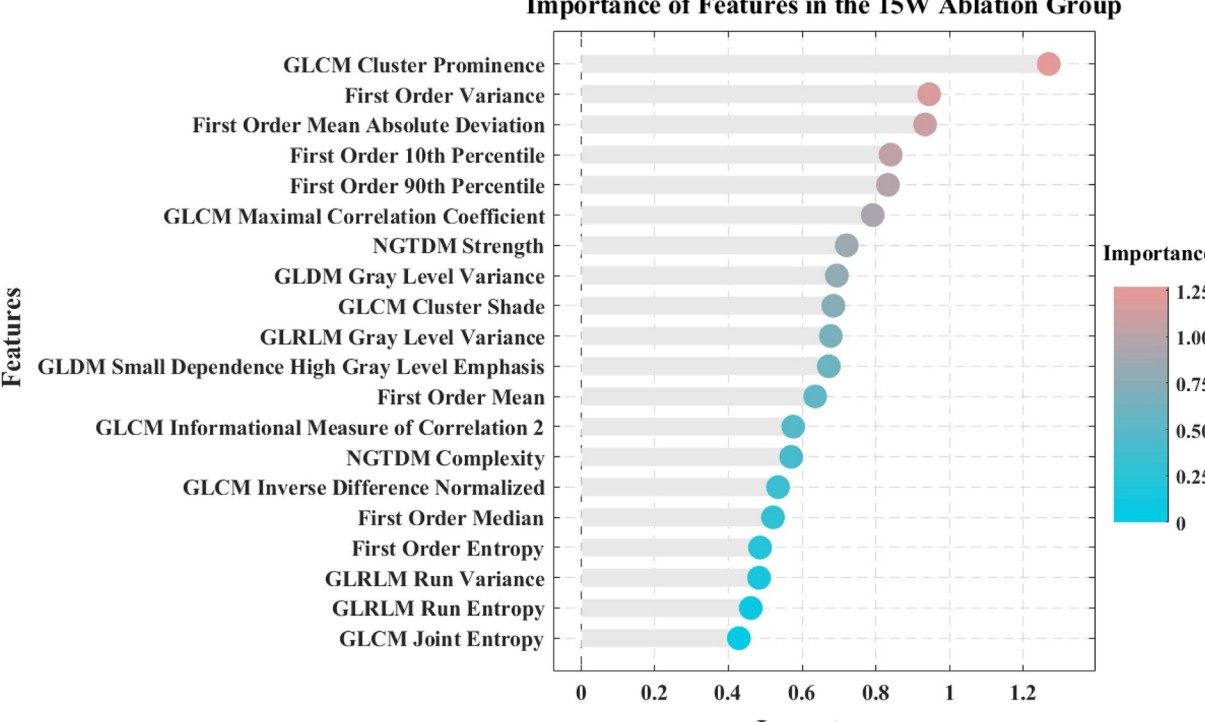

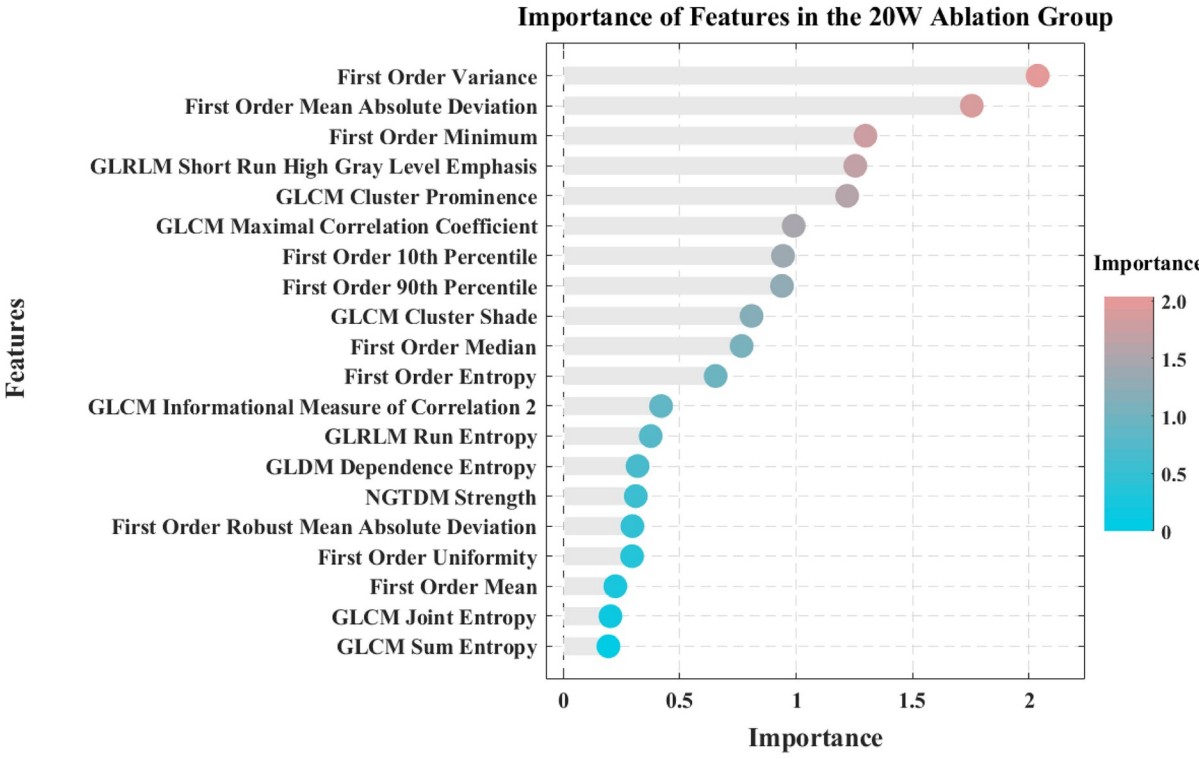

**Fig 5. Ranking of features by importance in the 15 W and 20 W ablation groups.** This figure displays the importance ranking of temperature prediction features identified by the random forest algorithm, based on their contribution to predicting temperature changes. It highlights the features with the greatest predictive value during the microwave ablation process.

ultrasound image sample pairs for each case. The samples were randomly divided into training, validation, and test sets in a 4:1:1 ratio. By training the ultrasound temperature prediction algorithm on classified samples, we have successfully established a temperature prediction model. This model can be utilized for subsequent temperature forecasting tasks.

## 4. Experimental results

### 4.1 Evaluation metrics for the prediction model

The temperature prediction model is used for single-point temperature prediction and thermal map prediction. In single-point temperature prediction, the model predicts the temperature at the center of the region of interest (ROI) corresponding to the measured point and compares it with the actual temperature obtained by the thermometer. In thermal map prediction, the model predicts the temperature of each pixel in the plane of the microwave ablation center area at the end of ablation, then stitches all the pixels together to present a temperature distribution map. In this study, the evaluation metrics for single-point temperature prediction are mean absolute error (MAE) and root mean square error (RMSE), while the evaluation metric for thermal map prediction is the error ratio.

①Mean Absolute Error (MAE) for single-point temperature prediction:

$$MAE(MeanAbsoluteError) = \frac{1}{n}\sum_{i=1}^{n}|y_i - \hat{y}_i|$$

Where n is the total number of temperature prediction samples, $y_i$ is the actual temperature value of sample i, and $\hat{y}_i$ is the predicted temperature value of sample i.

②Root Mean Square Error (RMSE) for single-point temperature prediction:

$$RMSE(Root\ Mean\ Square\ Error) = \sqrt{\frac{\sum_{i=1}^{n}(y_i - \hat{y}_i)^2}{n}}$$

where n is the total number of temperature prediction samples, $y_i$ is the actual temperature value of sample i, and $\hat{y}_i$ is the predicted temperature value of sample i.

③Error Ratio (ER) at the end of ablation for pixel area threshold and clinical observation area:

This study precisely calculated the clinical observation area $S_v$ delineated by the hyperechoic regions in ultrasound images after ablation, as well as the actual coagulation zone area $S_r$ outlined by the brownish-yellow parenchymal region in porcine liver post-ablation. Accurate quantitative analyses of these two evaluation metrics were achieved by integrating ultrasound images (S3 Fig) with gross specimen photographs (S1 Fig). Additionally, temperature distribution heat map predictions (S2 Fig) were made for specified regions based on the ablation center in ultrasound images. First, pixel blocks in the target area were extracted, and a regression model was constructed using the radiomic features of the extracted pixel blocks to predict the temperature. Then, based on a set temperature threshold, areas in the heat map with temperatures above the threshold were segmented and their areas were calculated to obtain the pixel area $S_p$. By separately comparing $S_v$ and $S_p$ with $S_r$, it is possible to calculate the error ratio (ER) of two different prediction methods. The closer ER is to 0, the smaller the prediction error of the coagulation zone area for the respective prediction method.

$$ER(Error\ Ratio) = \frac{|x - S_r|}{S_r}$$

In this equation, $x$ represents the measured area of the ablation zone ($S_v$) or the predicted area ($S_p$), as indicated in the corresponding analysis.

## 4.2 Analysis of temperature prediction results

In the experiment with an ablation power of 15 W, the temperature prediction of the validation set showed that the model's MAE was 2.84˚C, and the RMSE was 4.74˚C. The temperature prediction of the test set further demonstrated the accuracy of the model, with an MAE reaching 1.14˚C. Three samples involving a larger temperature range were selected from the experimental dataset for the analysis of the entire ablation process temperature prediction (Fig 6). The MAE of temperature measuring point 1 was 1.71˚C, that of temperature measuring point 2 was 2.08˚C, and that of temperature measuring point 3 was 2.14˚C. The experimental prediction results indicate that for the specific temperature measuring points mentioned above, the temperature prediction shows a close trend. However, there is a phenomenon where the

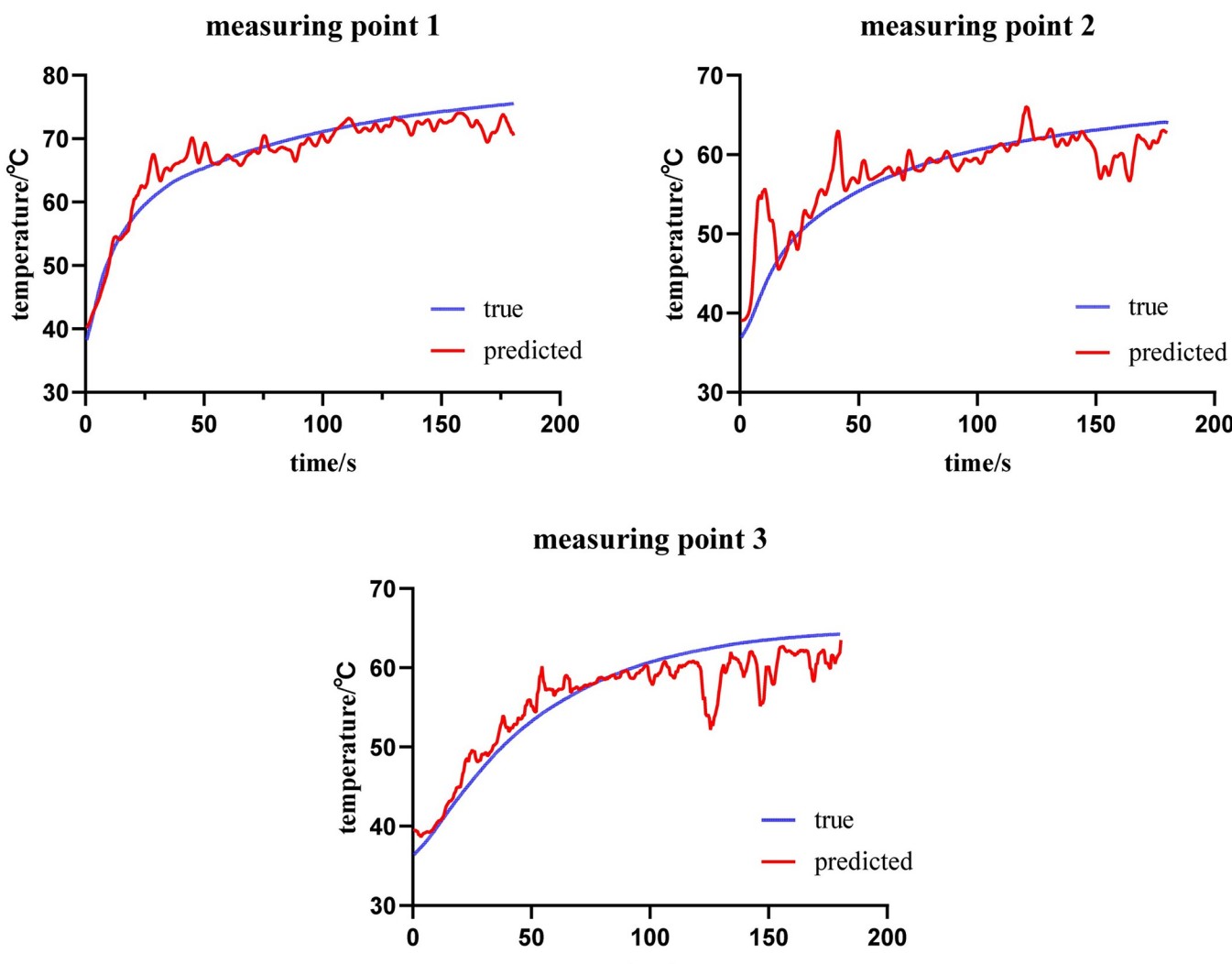

**Fig 6. Comparison of predicted and actual temperatures at 15W ablation (data presented in S1 Table).**

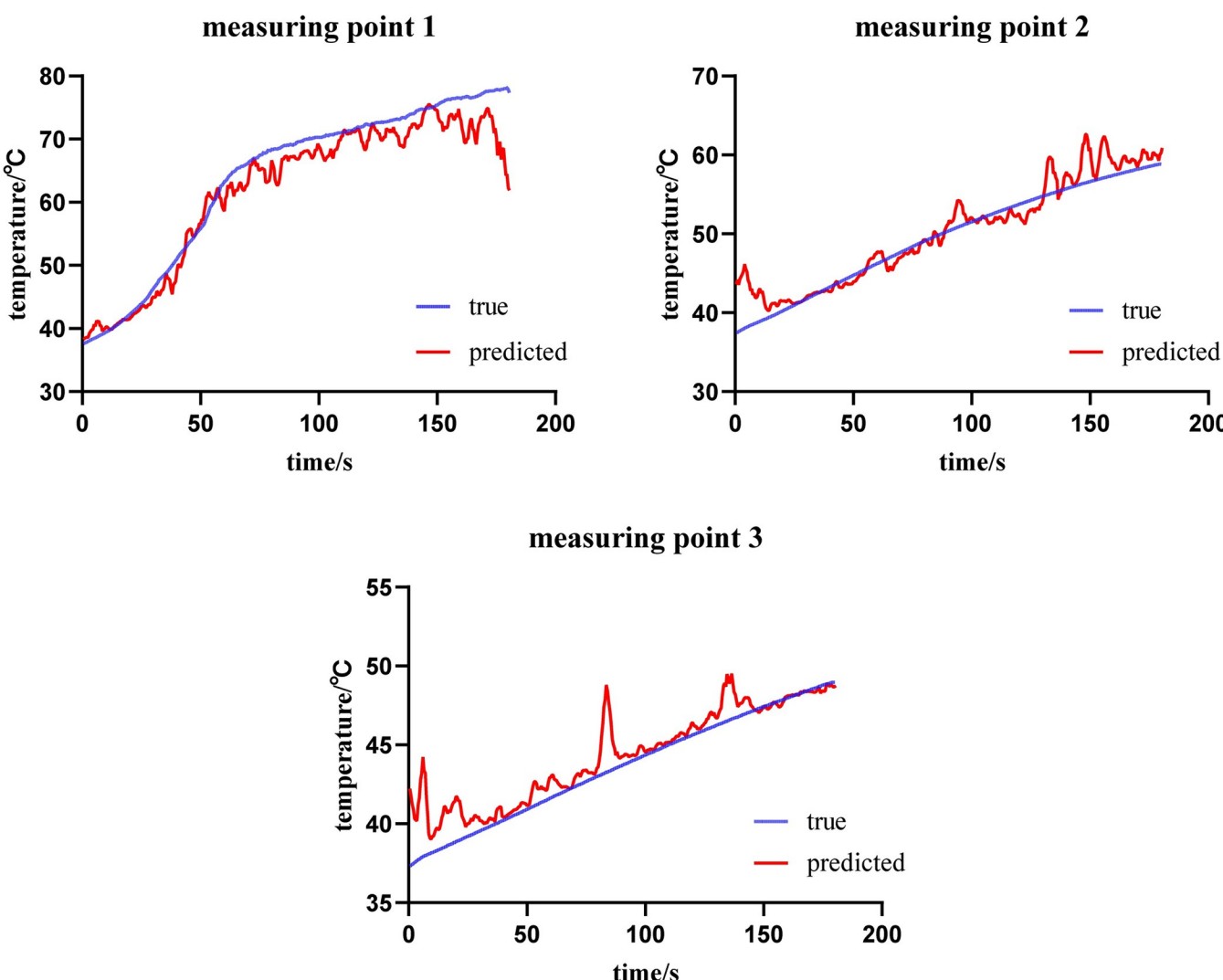

**Fig 7. Comparison of predicted and actual temperatures at 20W ablation (data presented in S2 Table).**

prediction error between the predicted temperature and the actual temperature is larger in the early and late stages of ablation.

In the experiment with an ablation power of 20 W, the MAE and RMSE of the validation set were 5.22°C and 7.61°C, respectively, showing an increase in error compared to the 15 W power group, which may be related to the overall increase in the temperature range involved. The MAE of the test set decreased to 4.73°C, indicating that the model still possesses good predictive accuracy at this power level. Three samples involving different temperature ranges were selected from the experimental dataset for the analysis of the entire ablation process temperature prediction (Fig 7). The MAE of measuring point 1 was 2.44°C, the MAE of measuring point 2 was 1.44°C, and the MAE of measuring point 3 was 0.89°C. The experimental prediction results indicate that for measuring point 1 in the 20 W power group, the overall prediction error is higher due to the larger range of prediction, as the maximum temperature reached over 75°C. In contrast, measuring points 2 and 3 had lower maximum temperatures, resulting in smaller prediction errors.

### 4.3 Analysis of predicted results in the coagulation zone

Representative ex vivo porcine liver ultrasound grayscale images, machine learning-predicted heatmaps, threshold heatmaps, and tissue section images collected immediately after MWA are presented in Fig 8. These images are divided into two groups based on the power used: 15W (Fig 8A–8D) and 20W (Fig 8E–8H). The threshold heatmaps display pixels with temperatures above 54˚C and calculate their area. In clinical ablation procedures, temperatures above 54˚C can ensure complete necrosis of tumor cells [5] and are close to the temperature at the ablation boundary. After ablation, the liver tissue is cut along the microwave ablation needle tract, and the clearly demarcated brownish-yellow area around the needle tract center is identified as the ablation coagulation range (Fig 8D and 8H). The coagulation area is outlined and measured using ImageJ software to obtain the actual coagulation area, which serves as the gold standard.

During ablation, an elliptical high-echo hyperechoic area centered around the ablation needle gradually expands on the ultrasound image, caused by microbubbles formed due to the temperature increase. The major and half the minor diameters (a, b) of the hyperechoic area's largest cross-section are measured immediately after ablation (Fig 9), and the area of the cavitation range is calculated using the formula for an ellipse: $S_v = \frac{1}{2} \times a \times b \times \pi$

In our study, Tables 1 and 2 are specifically designed to evaluate the accuracy of machine learning (ML) methods in predicting the coagulation zone. These tables compare the area of the coagulation zone predicted by ML $S_p$ with the gold standard area $S_r$ obtained using ImageJ software, and also include the area of the hyperechoic area in ultrasound grayscale images $S_v$. We selected samples from the 15W and 20W ablation groups and predicted the coagulation zone using the ML method based on the 54˚C isotherm. This threshold is close to the temperature at the ablation boundary and represents the minimum temperature required for protein denaturation, as determined in previous studies. From the data in Tables 1 and 2, we observe that the mean error ratios for the ML measurements in the 15W and 20W ablation groups are 0.159 and 0.122, respectively. These values are significantly lower than the mean error ratios for the hyperechoic area measurements in ultrasound grayscale images, which are 0.402 and

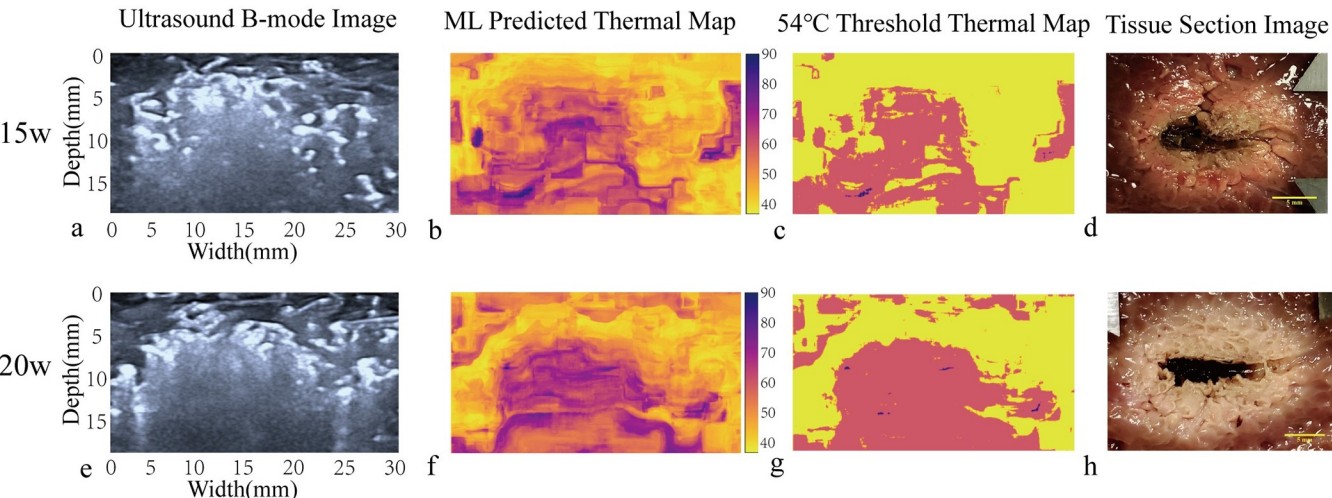

**Fig 8. Ultrasound images and corresponding predicted heat maps at the end of ablation.** This figure includes ex vivo porcine liver grayscale ultrasound images, machine learning-predicted thermographs, threshold thermographs, and tissue cross-sections. All images are categorized by ablation power, with the 15 W and 20 W groups displayed side by side for comparison.

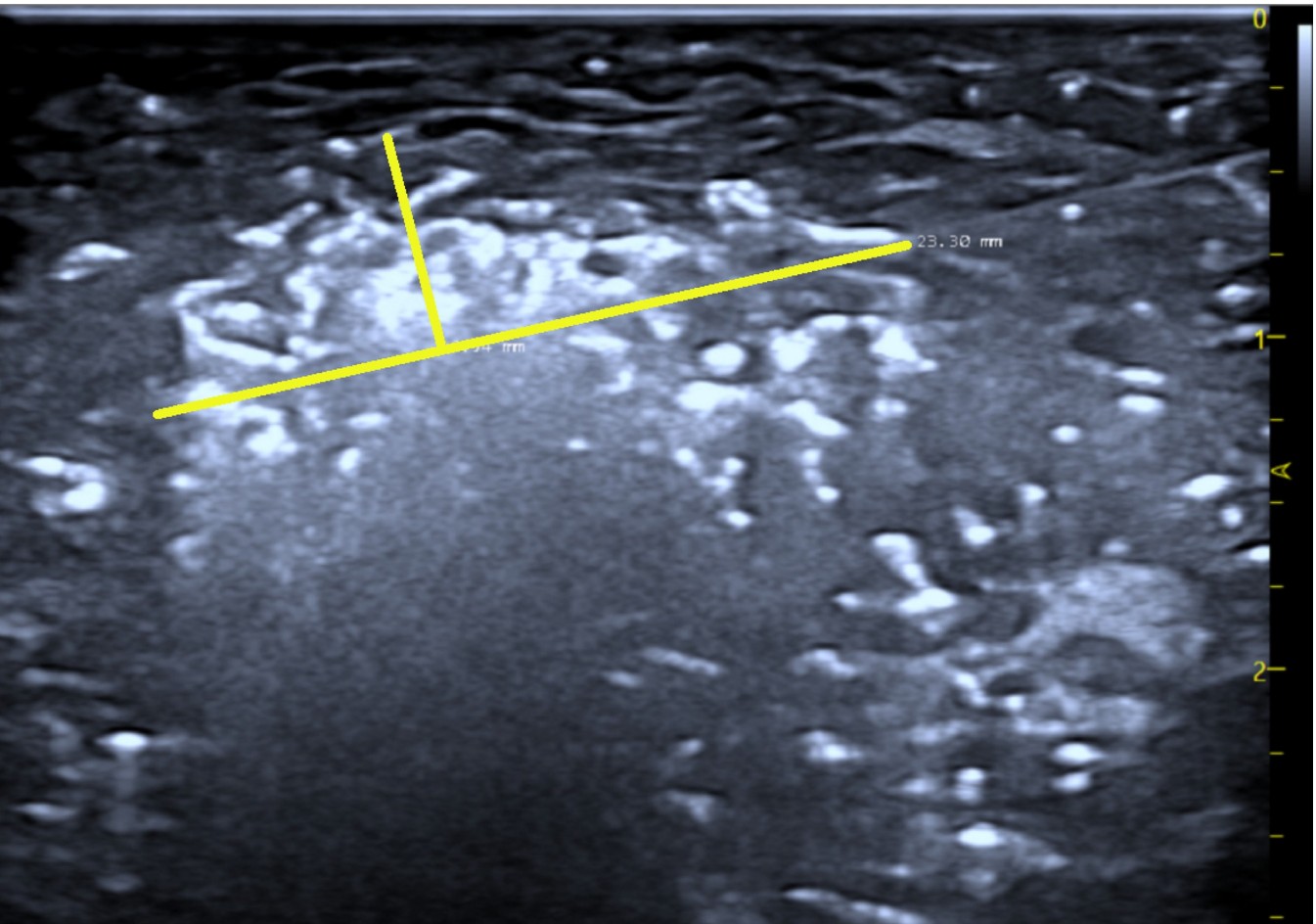

**Fig 9. Ultrasound hyperechoic area measurement.**

0.182, respectively. This demonstrates the superior accuracy of the ML method in predicting the coagulation zone.

## 4.4 Model error and correlation analysis of texture features with temperature

Through feature selection on the experimental data of 42 cases with an ablation power of 15 W, the top three features ranked by importance are Gray Level Co-occurrence Matrix

**Table 1. Prediction results of the coagulation area ($P$ = 15W, $t$ = 180s).**

| Gold Standard (mm²) | Measured Hyperechoic Area (mm²) | Predicted by ML (mm²) | Error Ratio | |
|---|---|---|---|---|
| | | | Hyperechoic Area | ML |
| 192.969 | 258.423 | 133.690 | 0.339 | 0.307 |
| 153.100 | 221.558 | 146.042 | 0.447 | 0.046 |
| 176.083 | 233.870 | 144.782 | 0.328 | 0.178 |
| 152.061 | 218.191 | 178.112 | 0.435 | 0.171 |
| 169.047 | 246.556 | 153.136 | 0.459 | 0.094 |

**Table 2. Prediction results of the coagulation area ($P$ = 20W, $t$ = 180s).**

| Gold Standard (mm²) | Measured Hyperechoic Area (mm²) | Predicted by ML (mm²) | Error Ratio | |
|---|---|---|---|---|
| | | | Hyperechoic Area | ML |
| 193.357 | 236.522 | 188.719 | 0.223 | 0.024 |
| 261.281 | 308.494 | 205.808 | 0.181 | 0.212 |
| 256.924 | 301.556 | 207.278 | 0.174 | 0.193 |
| 241.527 | 281.911 | 217.388 | 0.167 | 0.100 |
| 264.442 | 307.946 | 242.652 | 0.165 | 0.082 |

(GLCM) Cluster Prominence, First Order Variance, and First Order Mean Absolute Deviation. Among these, the Cluster Prominence feature is the most important in the 15 W random forest model, with an importance value of 1.2692. The importance rankings from the random forest model indicate the correlation between these three features and temperature, hence they were chosen as representative features and subjected to regression analysis with the corresponding temperature data. We selected a case from the 15 W dataset with good temperature prediction results. Fig 10 shows the fitting results of the three representative features with temperature in this dataset, with each scatter plot containing 2520 data points. The linear correlation coefficients between Cluster Prominence, Variance, and Mean Absolute Deviation with temperature are -0.8742, -0.7573, and -0.6760, respectively.

Through feature selection on the experimental data of 41 cases with an ablation power of 20 W, the top three features ranked by importance are First Order Variance, First Order Mean Absolute Deviation, and First Order Minimum. Among these, Variance is the most important feature in the 20 W random forest model, with an importance value of 2.0368. A case from the 20 W dataset with good temperature prediction results was selected, and these three representative features were subjected to regression analysis with temperature. Fig 11 shows the fitting results of the three features with temperature in this dataset, with each scatter plot containing 2520 data points. The linear correlation coefficients between Variance, Mean Absolute Deviation, and Minimum with temperature are -0.9016, -0.9061, and 0.7535, respectively.

## 5. Discussion

To address the challenge of determining the coagulation zone boundaries during clinical ablation procedures, we proposed a temperature monitoring method under ultrasound imaging based on supervised machine learning. The results indicate that this method can more effectively assess the coagulation zone compared to the commonly used clinical practice of evaluating ablation status based on grayscale ultrasound images. Our study addresses several important issues.

Compared to RF signals, grayscale signals are more convenient to acquire. RF signal acquisition typically requires research-grade ultrasound scanners, whereas grayscale signals can be collected using clinically used ultrasound devices. Texture features are obtained from grayscale signals, and many studies have shown a correlation between texture parameters and temperature [22, 28, 29]. Alvarenga et al. [22] analyzed temperature changes using certain parameters (entropy and correlation) of the gray level co-occurrence matrix (GLCM) by heating tissue phantoms in a water bath, showing that these parameters are highly sensitive at low temperatures. Yang et al. [28] studied the relationship between texture features and tissue temperature during microwave ablation, finding that GLCM parameters (contrast, angular second moment, inverse difference moment, and correlation) showed high correlation over a broader temperature range. Finally, Wang et al. [29] expanded this field by investigating texture

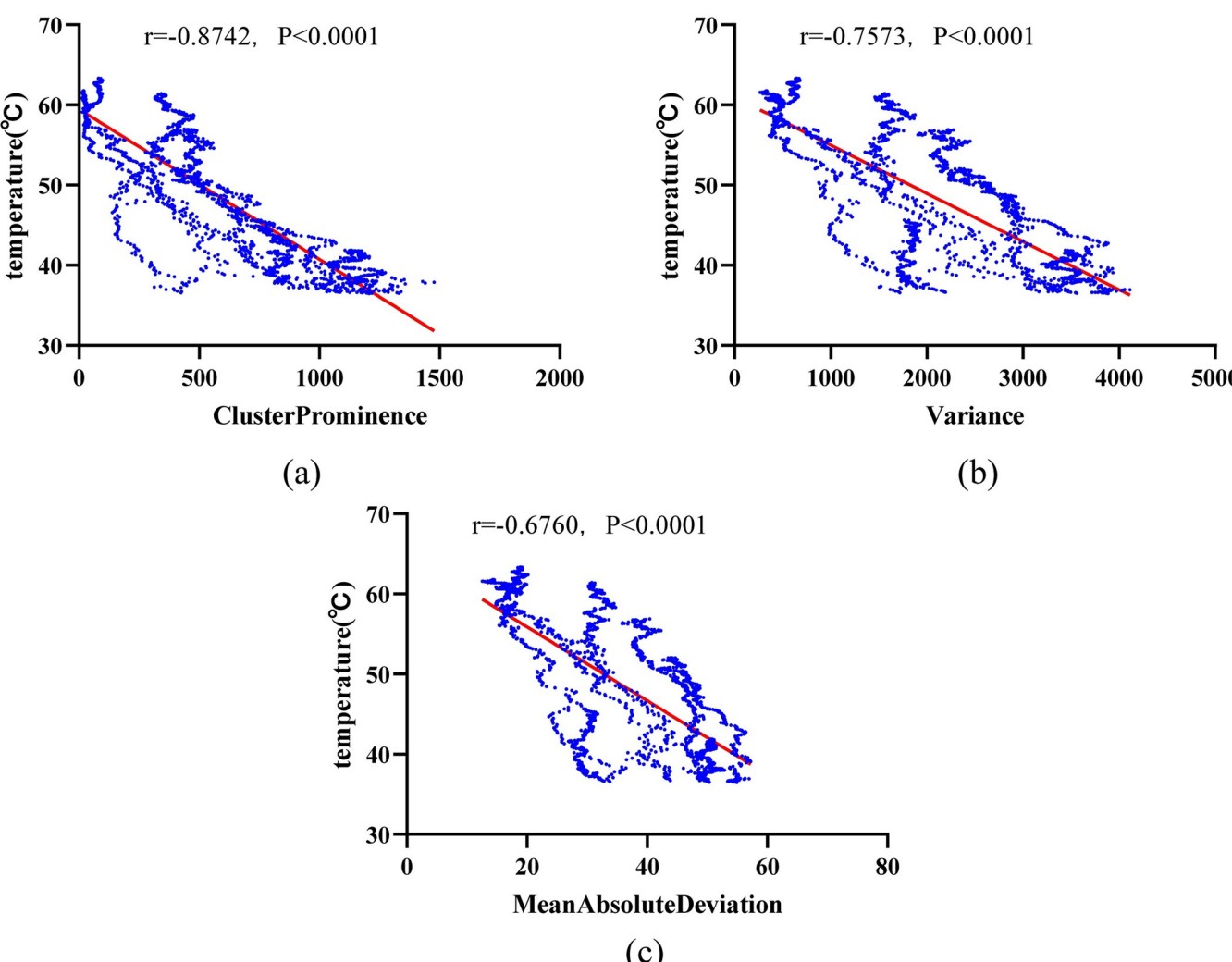

**Fig 10. Scatter plots of representative features with temperature in the 15 W ablation group.** (a) Cluster Prominence; (b) Variance; (c) Mean Absolute Deviation.

features based on gray level histograms, GLCM, and gray level gradient co-occurrence matrix in heated porcine liver samples, demonstrating that the correlation coefficients of five feature parameters exceeded 0.9 in the temperature range of 20°C to 60°C. Our study comprehensively evaluated texture parameters by screening 104 feature values, obtaining features with high correlation to temperature for model testing. Not only can the trends of feature values with temperature changes be visually observed through scatter plots, but the selected texture features also have interpretability. In our study, although the important features identified for temperature prediction varied across different power groups, we observed a clear consistent trend: as temperature increased, the grayscale values of the ablated tissue also increased, leading to greater image uniformity. Notably, features such as GLCM cluster prominence, variance, mean absolute deviation, and minimum value exhibited high importance in both ablation groups, playing crucial roles in temperature prediction. Correlation analysis of these features with temperature revealed significant patterns: the minimum value was positively correlated with temperature, indicating a general increase in grayscale values within the region of interest (ROI) as temperature rose. Conversely, GLCM cluster prominence, variance, and

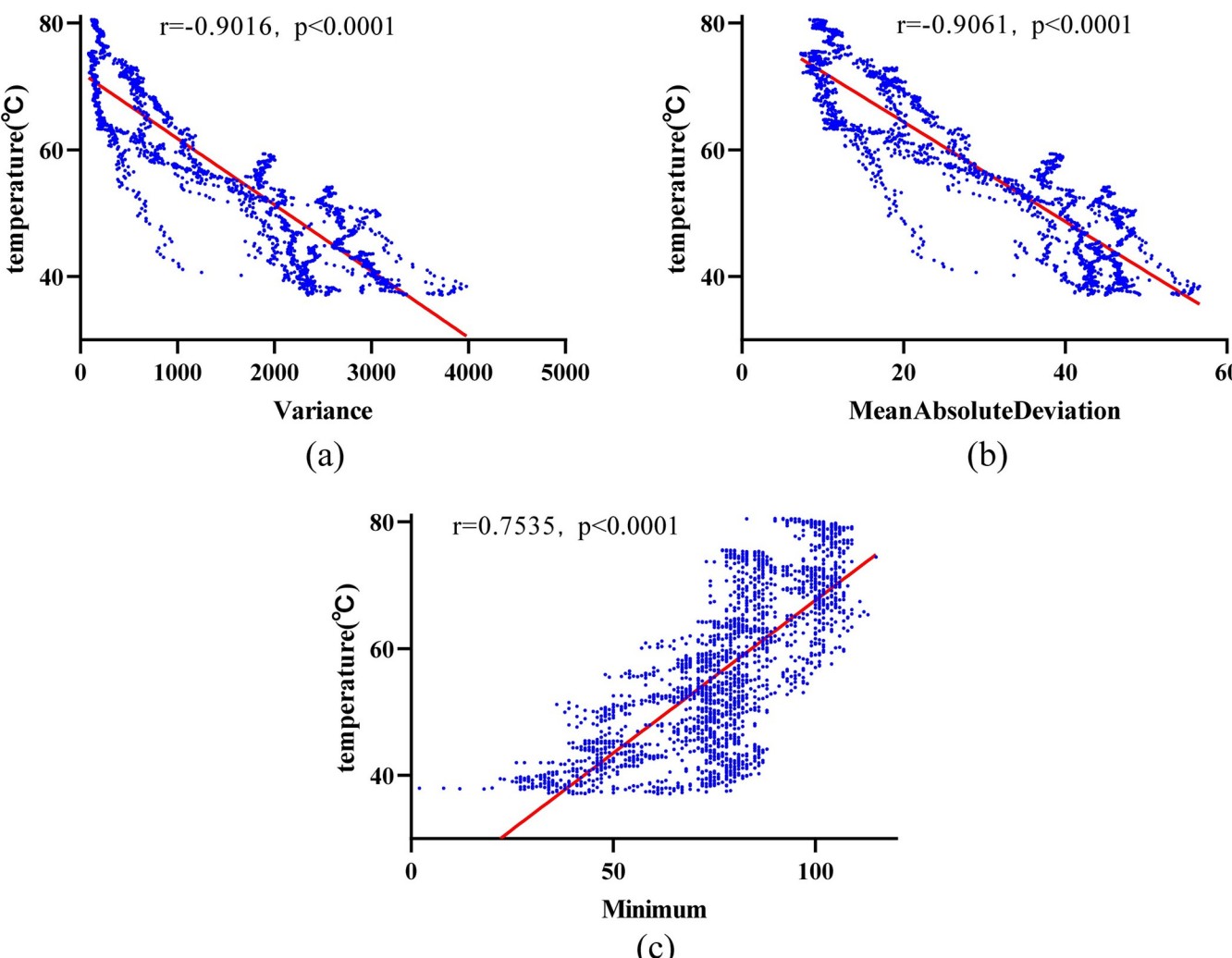

**Fig 11. Scatter plots of representative features with temperature in the 20 W ablation group.** (a) Variance; (b) Mean Absolute Deviation; (c) Minimum Feature Value.

mean absolute deviation were negatively correlated with temperature. As measures of skewness and asymmetry, these features decreased with rising temperature, reflecting increased uniformity in grayscale images due to tissue coagulation and bubble formation. Despite potential variations in temperature-related features identified by different studies due to differences in experimental conditions and tissue characteristics, we believe the observed pattern of grayscale image changes with temperature is universal. This conclusion holds clinical significance across various contexts, offering a reliable method for predicting and assessing temperature changes in coagulation zone.

For machine learning, achieving ideal results often requires a large amount of data for training. Traditional ablation experiments usually use a single temperature-measuring needle parallel to the ablation needle [13, 30], collecting data from only one position per ablation. In contrast, our study innovatively used a custom mold, allowing the thermocouples to be evenly distributed in a plane perpendicular to the ablation needle. A single ablation experiment can simultaneously collect temperature data from seven positions, improving data collection efficiency and making sampling in different positions of the ablation area more representative.

Our experiment achieved good results in temperature prediction, with mean absolute errors within 5˚C, and good prediction results for temperatures above the critical temperature. Additionally, by predicting the temperature of all pixels in the selected coagulation zone, we obtained the corresponding heatmaps, which better illustrate the spatial temperature distribution in the entire ultrasound imaging area and closely match the gross gold standard in predicted area. In the experimental prediction of single-point temperatures, we observed a deviation between predicted and actual values, with errors primarily concentrated at the initiation and termination phases of ablation. This deviation may stem from the rapid temperature changes during the initial phase of ablation, leading to a sharp increase in the number of bubbles, as well as reduced image variation due to decreased bubble aggregation at the end of ablation, partially masking texture information of underlying tissues. Additionally, as ablation power and temperature increase, prediction errors correspondingly escalate. During the experiment, we observed that the hyperechoic region was generally larger than the coagulation zone, and the gap between the two gradually decreased as ablation time was extended. This observation is consistent with the findings of Uehara T et al. [31], who introduced the concept of the hyperechoic margin, which aligns with what we observed. Additionally, we noted that the machine learning (ML) method's predictions of coagulation zone areas were typically smaller than the gold standard data delineated from ex vivo tissues. We speculate that this discrepancy may arise from the ML method's limitations in recognizing the coagulation zone's edges during the image segmentation process. The ML method predicts the area by summing the temperatures of all pixel points above 54˚C during the image segmentation process. While this threshold is close to the temperature at the ablation boundary, the inherent error in temperature prediction may result in an underestimation of boundary temperatures. Especially, the scarcity of temperature data around 54˚C during data collection might further affect the ML method's predictive accuracy for this critical temperature range. In the temperature prediction heat maps, we noted abnormally elevated temperatures in certain areas, which, in reality, were at a distance from the coagulation zone. This could be attributed to bubble diffusion in vessels and fascia, potentially causing misinterpretation of ablation efficacy. Thus, there remains potential for improvement in the accuracy of our temperature prediction and heat map models.

Our experiment has some limitations. Firstly, using thermocouples as the gold standard for temperature measurement means that even if the prediction results are close to the thermocouple measurements, they only represent the temperature at the corresponding spatial position. Evaluating the temperature of the entire coagulation zone may require magnetic resonance imaging. Furthermore, there exists a disparity between the ablation method employed in our experiments and the actual conditions of microwave ablation for liver cancer in clinical practice. In clinical settings, microwave ablation therapy for liver cancer typically utilizes power ranging from 60 W to 100 W for heating, a range considered suitable for ablation [32]. However, in our experimental research, due to the limited thickness of porcine liver samples and the absence of heat dissipation effects from blood flow in the in vivo environment, we adjusted the ablation strategy based on the requirements of data collection from our experiments to ensure a more extensive temperature distribution and prevent the diffusion of ablation edges to the liver capsule surface. This adjustment involved lowering the ablation power and extending the heating time to 180 seconds to achieve precise control over the ablation effect. Lastly, our experiments were conducted on ex vivo porcine liver samples, whereas in actual ablation surgeries for clinical patients, considerations must be made for the effects of blood flow and respiratory motion, which are factors that cannot be overlooked. These influencing factors are also the primary reasons why various temperature measurement techniques have not yet been successfully applied in clinical ablation surgeries.

## 6. Conclusion

This study proposed a predictive model for tissue temperature during microwave ablation therapy. We used the random forest method to select and train ultrasound grayscale texture features to evaluate tissue temperature, providing guidance for monitoring the coagulation zone of liver tumor microwave ablation.

## Supporting information

**S1 Table. Predicted and measured temperatures for the 15 W power group.**
(XLSX)

**S2 Table. Predicted and measured temperatures for the 20 W power group.**
(XLSX)

**S1 Fig. Cross-sectional photos of liver post-ablation for 15 W and 20 W power groups.**
(PDF)

**S2 Fig. Thermal maps for 15 W and 20 W power groups: Machine learning predictions and 54˚C threshold highlights.**
(PDF)

**S3 Fig. Ultrasound greyscale images at the end of ablation for 15 W and 20 W power groups.**
(PDF)

## Author Contributions

**Conceptualization:** Yan Huang, Zhiyong Zhou.

**Data curation:** Yan Xiong, Yi Zheng.

**Formal analysis:** Yan Xiong, Yi Zheng, Yuxin Wang.

**Funding acquisition:** Yan Xiong, Zhiyong Zhou.

**Investigation:** Yan Xiong, Yi Zheng, Wei Long, Qin Wang, Yi You, Yuheng Zhou, Jiang Zhong, Yunxi Ge, Youchen Li, Yan Huang, Zhiyong Zhou.

**Methodology:** Yan Xiong, Yi Zheng, Yan Huang, Zhiyong Zhou.

**Project administration:** Yan Xiong, Yi Zheng, Zhiyong Zhou.

**Software:** Yi Zheng, Yuxin Wang, Zhiyong Zhou.

**Supervision:** Yan Huang.

**Validation:** Yan Xiong, Wei Long.

**Visualization:** Yan Xiong, Yi Zheng.

**Writing – original draft:** Yan Xiong, Yi Zheng.

**Writing – review & editing:** Yan Xiong, Yi Zheng, Wei Long, Qin Wang, Yi You, Yuheng Zhou, Yan Huang, Zhiyong Zhou.

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
