## [Decision Letter · Decision Letter 0]

7 Jun 2024

PONE-D-24-15770Study on Microwave Ablation Temperature Prediction Model Based on Grayscale Ultrasound Texture and Machine LearningPLOS ONE

Dear Dr. Huang,

Thank you for submitting your manuscript to PLOS ONE. After careful consideration, we feel that it has merit but does not fully meet PLOS ONE’s publication criteria as it currently stands. Therefore, we invite you to submit a revised version of the manuscript that addresses the points raised during the review process.

We look forward to receiving your revised manuscript.

Kind regards,

Carlos Alberto Antunes Viegas, DVM; MSc; PhD

Academic Editor

PLOS ONE

“This work was supported partly by National Natural Science Foundation of China (62271480), Youth Innovation Promotion Association CAS (2021324), Jiangsu Key Technology Research Development Program (BE2021612), Science and Technology Development Project of Suzhou (SYG202321), Postgraduate Research and Practice Innovation Program of Jiangsu Province (KYCX23_2093).”

“This work was supported partly by National Natural Science Foundation of China (62271480), Youth Innovation Promotion Association CAS (2021324), Jiangsu Key Technology Research Development Program (BE2021612), Science and Technology Development Project of Suzhou (SYG202321), Postgraduate Research and Practice Innovation Program of Jiangsu Province (KYCX23_2093). The funders had no role in study design, data collection and analysis, decision to publish, or preparation of the manuscript.”

5. We are unable to open your Supporting Information file [S3 Fig.rar,S4 Fig.rar and S5 Fig.rar]. Please kindly revise as necessary and re-upload.

Reviewers' comments:

Reviewer's Responses to Questions

**Comments to the Author**

1. Is the manuscript technically sound, and do the data support the conclusions?

Reviewer #1: Yes

Reviewer #2: No

2. Has the statistical analysis been performed appropriately and rigorously? 

Reviewer #1: Yes

Reviewer #2: Yes

3. Have the authors made all data underlying the findings in their manuscript fully available?

Reviewer #1: Yes

Reviewer #2: Yes

4. Is the manuscript presented in an intelligible fashion and written in standard English?

Reviewer #1: Yes

Reviewer #2: No

5. Review Comments to the Author

Reviewer #1: 1.Is the selection of temperature measurement points related to the subsequent machine learning model? Why not select the vertical cross-section of the antenna as the temperature measurement point to predict the ablation radius? What is the necessity of vertically arranging 7 points? What is the significance of different distances between temperature measurement points in Figure 1?

2.On line 264, does Sv represent the cavitation area? It's best to have a unified statement

3.On line 278, x represents whether it represents Sv or Sp respectively. It is suggested to clarify the meaning of x

4.Comparing Figure 6 with Figure 7, it should be that the temperature is low, but there is an abnormal phenomenon at temperature measurement points 2 # and 3 #. How to explain this?

5.In the analysis of the results on line 305, did we conclude that the larger the prediction range, the greater the error? Or is it just a case study?

6.Figures 10 and 11 show Scatter plots of representative features with temperature at different power levels, with different conclusions. Therefore, whether there is a certain pattern for the important features commonly used in clinical microwave power levels of 60-100W or different heating times, or whether all the processes in the article need to be repeated, please provide the universality and significance of this study in the main text

7.There is too little introduction to the content of machine learning, which makes it difficult to fully understand the author's model, especially regarding how the first three features of importance ranking in section 4.4 were obtained?

Reviewer #2: General Comments:

MRI and CT treatment monitoring currently offer the greatest potential for thermal therapies to emerge into clinical practice. That said, given the associated high cost and limited accessibility, there is a need for cost-effective treatment monitoring tools like what is presented in this manuscript. Overall the temperature prediction method and results are interesting, but the image analysis results are somewhat confusing. A number of deficiencies need to be addressed.

First of all, it is not clear what treatment outcomes are being measured and compared in Fig 8. Throughout the manuscript the authors refer to ablation zone, vaporization zone, coagulation zone and cavitation area. The authors need to clearly state the tissue characteristics for these regions (what is going on) and if these regions of the thermal lesion are distinct or overlapping. It is clear that at the end of the heating (Fig. 8d,h) there is a region of tissue charring at the ablation needle surrounded by tissue coagulation. Without some description of the mechanisms for thermal lesion formation, it is difficult to interpret the data and analysis. For example, it is not clear in Tables 1 and 2 which data sets are to be compared. The microbubble zone is consistently larger than the tissue coagulation zone, but is this expected? In addition, the ML method consistently underestimates the coagulation area compared to the ImageJ data (used as a gold standard), but no explanation is provided. This needs to be discussed.

I'd also recommend to not refer to the formation of microbubbles as cavitation, as the mechanism here is MW heating, and is different from US induced cavitation. I found this confusing.

The images Fig 8d,h, clearly show tissue charring (temperatures in excess of 100 C, leading to water vaporization - microbubble formation). This is typically to be avoided during thermal therapy, due to the associated smoke production and the resulting change in tissue permittivity which alters antennae performance and power delivery. So a better test would be using MW powers that result in a maximum tissue temperature of 90 C.

In the Discussion (p.10), the author's state "The results indicate that this method can more effectively assess the ablation area.", but offer no justification, i.e., compared to what other method(s).

Specific Comments/Questions:

1. p.4, It is not clear what is meant by "texture information changes", please add a brief explanation.

2. What is the US frequency? The authors refer to a sampling frequency of 15 MHz, but that is not US frequency.

3. p.6, What type of thermocouple is used, Type-K, Type-T?

4. p.6, What are the dimensions of the mold (e.g what volume of liver is used in each experiment).

5. Fig 1B, It appears that the thermocouples are all not in the same plane. Please clarify. I'd also suggest changing in the caption “theoretical” to “desired” and include text to describe the temperature probes (blue) and ablation needle (red).

6. Fig 3 caption needs to be expanded to include a description of the individual process steps.

7. Fig 4 caption needs to be expanded to describe the image, e.g. An US frame showing the ablation needle and……..

8. Fig 5 caption needs to be more descriptive.

9. Fig 8 image, “width” not “wideth” and the caption needs to be more descriptive.

10. p. 10, The radiometric texture features used in the analysis are only referenced. These need to be described at least briefly in the manuscript for the reader to understand their utility.

6. PLOS authors have the option to publish the peer review history of their article (what does this mean?). If published, this will include your full peer review and any attached files.

Reviewer #1: No

Reviewer #2: No

---

## [Author Response · Author response to Decision Letter 0]

21 Jul 2024

Dear Editors,

We appreciate the opportunity to revise our manuscript entitled "Fast Ultrasonic Ablation Monitoring: An Innovative Approach Using Ultrasound RF Signals and Singular Value Decomposition" for consideration at PLOS ONE. We have carefully considered the reviewers' comments and have made the following revisions to our manuscript:

1. Style and File Naming: We have revised our manuscript according to PLOS ONE's style requirements, including the naming of files. We have ensured that all documents adhere to the guidelines provided in the style templates. Should there be any discrepancies we may have overlooked, we kindly ask for your guidance.

2. Acknowledgments and Funding Statement: We sincerely apologize for the oversight regarding the placement of funding information in the Acknowledgments section. We have removed the funding details from this section in our revised manuscript. We request that the following statement be retained in the Funding Statement section of the online submission form: "This work was supported partly by National Natural Science Foundation of China (62271480), Youth Innovation Promotion Association CAS (2021324), Jiangsu Key Technology Research Development Program (BE2021612), Science and Technology Development Project of Suzhou (SYG202321), Postgraduate Research and Practice Innovation Program of Jiangsu Province (KYCX23_2093). The funders had no role in study design, data collection and analysis, decision to publish, or preparation of the manuscript."

3. Data Sharing: After thorough discussion, we have decided to make our data publicly available. The primary data related to our study have been uploaded as Supporting Information. Given the large volume of our original experimental data, which may not be suitable for deposition in a public database, we have provided access to this data through our submission. We welcome any inquiries for further experimental details via email correspondence.

4. ORCID iD: We acknowledge the requirement for the corresponding author to have an ORCID iD linked to the Editorial Manager account. We will ensure that this is done promptly.

5. Supporting Information File: We have converted the image groups from the compressed files [S3 Fig.rar, S4 Fig.rar, and S5 Fig.rar] into PDF format for ease of access and have re-uploaded them. We believe this will resolve the issue of file accessibility.

We trust that these revisions meet the journal's requirements and enhance the quality of our submission. We look forward to your positive response.

Sincerely,

Professor Yan Huang

Department of Ultrasound, Nanjing Hospital of Chinese Medicine Aﬃliated to Nanjing University of Chinese Medicine, Nanjing, China

E-mail: jacob6666@163.com

July 21, 2024

Dear Editor:

Thank you very much for the valuable comments on our manuscript, “Fast Ultrasonic Ablation Monitoring: An Innovative Approach Using Ultrasound RF Signals and Singular Value Decomposition”. We have made every effort to revise the manuscript carefully, in light of the reports from the reviewers and editors. 

We sincerely hope the revised manuscript could be considered for publication on PLOS ONE. Really appreciate your time and kind consideration. Detailed responses can be found as the follows.

Best Regards!

Yan Huang

Reviewer #1: 

1. Is the selection of temperature measurement points related to the subsequent machine learning model? Why not select the vertical cross-section of the antenna as the temperature measurement point to predict the ablation radius? What is the necessity of vertically arranging 7 points? What is the significance of different distances between temperature measurement points in Figure 1?

Response: Thank you for your question. Our selection of temperature measurement points is closely related to the machine learning model, aiming to enhance the model's ability to learn temperature variations within the ablation zone using data from different locations. Simulation images reveal the non-uniformity of temperature distribution during ablation, confirming the necessity of setting measurement points at various positions.

We chose the cross-section parallel to the ablation needle for temperature measurement points because this cross-section is commonly used in clinical practice and provides richer tissue information. Additionally, as long as the long axis of the ablation needle is centered in the ultrasound image, changing the probe direction will not affect the accuracy of the ablation radius measurement. In contrast, the vertical cross-section can lead to measurement deviations when the probe position changes. Simulation images show that the ablation area forms an ellipsoid in three-dimensional space, and the ultrasound probe can only scan one section of it, leading to the aforementioned issues.

Simulation 2D/3D diagram

ZHU Roujun, et al. "Design of Precision Treatment Planning System for Tumor Microwave Thermal Ablation Based on LabWindows,"（in Cineses）, Life Science Instruments, vol. 19, no. 01, 2021, pp. 68-74. 

Arranging the temperature measurement points vertically ensures they are in the same plane as the ablation needle in the ultrasound image. This helps us accurately determine the position of the temperature probes and ensures that the data collected from each measurement point corresponds to the correct ultrasound ROI.

The distances between the temperature measurement points are carefully designed to meet practical needs. To collect more temperature data and improve the model's accuracy, we aim to place as many measurement points as possible within a limited space. However, spacing them too closely would cause acoustic shadow interference between the needles, affecting grayscale signal collection. Therefore, the uniform spacing of the temperature measurement points in Figure 1B ensures an optimal number of measurement points while maintaining image quality.

2. On line 264, does Sv represent the cavitation area? It's best to have a unified statement.

Response: Sv represents the hyperechoic "cavitation area" observed on the ultrasound image. However, we have noticed that the article contains multiple concepts related to ablation and cavitation, which may confuse readers. To address this issue, we have conducted a thorough review and revision of the entire text, clearly distinguishing between the tissue effects caused by microwave heating and the cavitation phenomenon induced by ultrasound. We have unified the terminology to avoid ambiguity:

The terms "ablation zone," "ablation area," and "ablation range" have been consistently revised to "coagulation zone" or "coagulation area" to clearly indicate the extent of tissue coagulation following microwave ablation therapy.

The terms "vaporization zone" and "cavitation area" have been unified as "hyperechoic area" to describe the high-echo phenomenon observed in ultrasound images due to the formation of gas within the tissue.

3. On line 278, x represents whether it represents Sv or Sp respectively. It is suggested to clarify the meaning of x

Response: Thank you for your suggestion. We have clarified the meaning of x in the revised manuscript (Lines 335-336).

4. Comparing Figure 6 with Figure 7, it should be that the temperature is low, but there is an abnormal phenomenon at temperature measurement points 2 # and 3 #. How to explain this?

Response: The temperature measurement points selected in the 15 W power group represented by Figure 6 have a broader prediction range, and theoretically, their overall temperature should be lower than that of the 20 W power group. However, due to the different distances of the seven measurement points from the high-temperature center, the points closer to the high-temperature center will naturally record higher temperatures. Consequently, some measurement points in the 15 W power group, such as 2# and 3#, actually recorded higher temperatures than certain measurement points in the 20W power group.

5. In the analysis of the results on line 305, did we conclude that the larger the prediction range, the greater the error? Or is it just a case study?

Response: We did observe a trend where temperature measurement points with larger prediction ranges exhibited greater prediction errors. This conclusion is not merely a result of a case study but is derived from our comprehensive analysis of the entire dataset. During data processing, we noticed that when the temperature range at the measurement points increased, particularly when more high-temperature segments were included, the model's prediction error tended to increase. This phenomenon was observed in data from multiple measurement points and was not limited to isolated cases. We believe this is because, in the high-temperature range, the changes in tissue grayscale signals can be more complex, making accurate predictions more challenging for the model. Therefore, we conclude that as the temperature prediction range expands, especially when it includes more high-temperature data, the prediction error tends to increase, as reflected in the higher mean absolute error.

6. Figures 10 and 11 show Scatter plots of representative features with temperature at different power levels, with different conclusions. Therefore, whether there is a certain pattern for the important features commonly used in clinical microwave power levels of 60-100W or different heating times, or whether all the processes in the article need to be repeated, please provide the universality and significance of this study in the main text.

Response: We recognize that different texture features may emerge at various clinical microwave power levels (60-100W) and with varying heating times. We agree that these features are influenced not only by power but also by tissue characteristics. Although there are differences between porcine liver and human liver tissue properties, which might yield different results, we believe the biological significance behind texture features remains consistent.

In our study, cluster prominence, variance, mean absolute deviation, and minimum were identified as highly important features in both ablation groups, indicating their key roles in temperature prediction. The correlation analysis of these features with temperature revealed the following patterns: the minimum value was positively correlated with temperature, while the other features showed a negative correlation. Specifically, an increase in the minimum value reflected a general rise in grayscale values within the ROI as temperature increased. In contrast, cluster prominence, variance, and mean absolute deviation, which measure skewness and asymmetry, decreased with rising temperatures, indicating that tissue coagulation and bubble formation made the grayscale images more homogeneous.

We understand that due to variations in experimental conditions and tissue characteristics, the features with high-temperature correlation might differ across studies. However, we believe one core conclusion is universal: as temperature increases, the grayscale values of ablated tissue increase, and the images become more homogeneous. For instance, in the literature cited in our discussion, Wang et al. observed that in ex vivo porcine liver heated in a water bath, texture feature parameters such as the mean grayscale of GLH, homogeneity of GLCM, hybrid entropy, inverse difference moment, and correlation of GGCM were closely related to temperature and increased with it. These changes in texture features suggest that similar patterns can be observed across different studies, supporting our conclusion.

To further investigate the universality of these features under different conditions, we plan to extend the power range and heating times in future studies, as well as include in vivo experiments to verify whether the above conclusions hold. We believe that through these additional studies, we can better understand the applicability of these features in various clinical scenarios.

We appreciate your suggestion and have incorporated the discussion on the universality and significance of the study into the revised manuscript in lines 491-509.

7. There is too little introduction to the content of machine learning, which makes it difficult to fully understand the author's model, especially regarding how the first three features of importance ranking in section 4.4 were obtained?

Response: In this study, feature importance was assessed using the Gini importance index from the random forest algorithm. The Gini importance index measures a feature's ability to reduce impurity at each split point within the random forest. A high Gini coefficient indicates that the feature provides more information during classification or regression, thus contributing more significantly to the model's predictive power. By calculating the Gini coefficient for each feature, we ranked their importance accordingly. In Section 4.4, we utilized machine learning methods to analyze the top three features based on their Gini coefficients. These features, identified as having the highest importance rankings, were subjected to detailed analysis. The rankings were consistent with those in Figure 5, which displays the importance of features for different power groups. These top features were identified due to their strong correlation with temperature changes, making them significant predictors in our model.

To provide a more comprehensive understanding of our model, we have supplemented the revised manuscript with additional details on the feature importance evaluation method (Lines 265-271).

Reviewer #2: MRI and CT treatment monitoring currently offer the greatest potential for thermal therapies to emerge into clinical practice. That said, given the associated high cost and limited accessibility, there is a need for cost-effective treatment monitoring tools like what is presented in this manuscript. Overall the temperature prediction method and results are interesting, but the image analysis results are somewhat confusing. A number of deficiencies need to be addressed.

General comments:

1. First of all, it is not clear what treatment outcomes are being measured and compared in Fig 8. Throughout the manuscript the authors refer to ablation zone, vaporization zone, coagulation zone and cavitation area. The authors need to clearly state the tissue characteristics for these regions (what is going on) and if these regions of the thermal lesion are distinct or overlapping. It is clear that at the end of the heating (Fig. 8d,h) there is a region of tissue charring at the ablation needle surrounded by tissue coagulation. Without some description of the mechanisms for thermal lesion formation, it is difficult to interpret the data and analysis. For example, it is not clear in Tables 1 and 2 which data sets are to be compared. The microbubble zone is consistently larger than the tissue coagulation zone, but is this expected? In addition, the ML method consistently underestimates the coagulation area compared to the ImageJ data (used as a gold standard), but no explanation is provided. This needs to be discussed.

Response: Thank you for your question. To enhance the clarity of our manuscript and eliminate conceptual confusion, we have thoroughly revised and clarified the key terms mentioned in the text. We have standardized the use of "coagulation zone" to refer to the tissue area where coagulation necrosis has occurred following microwave ablation. Additionally, we have unified "vaporization zone" and "cavitation area" under the term "hyperechoic area," which describes the region in the ultrasound grayscale image exhibiting high echogenicity due to gas formation within the tissue.

In our study:

The coagulation zone represents the area of protein denaturation and necrosis, which includes the charred region close to the ablation needle where temperatures are highest.

The hyperechoic area, observed in ultrasound images, is typically larger than the actual coagulation zone because it covers the region affected by gas bubbles formed during ablation.

To address the question regarding the comparison of data sets in Tables 1 and 2, we have included additional explanations in the revised manuscript (Lines 407-420). Tables 1 and 2 compare the coagulation zone areas

---

## [Decision Letter · Decision Letter 1]

5 Aug 2024

Study on Microwave Ablation Temperature Prediction Model Based on Grayscale Ultrasound Texture and Machine Learning

PONE-D-24-15770R1

Dear Dr. Yan Huang,

We’re pleased to inform you that your manuscript has been judged scientifically suitable for publication and will be formally accepted for publication once it meets all outstanding technical requirements.

Kind regards,

Carlos Alberto Antunes Viegas, DVM; MSc; PhD

Academic Editor

PLOS ONE

Additional Editor Comments (optional):

Dear authors

Your article is ready to be published, but you must consider the first reviewer's suggestion regarding the second round of revisions in the last version.

Best Regards

Reviewers' comments:

Reviewer's Responses to Questions

**Comments to the Author**

1. If the authors have adequately addressed your comments raised in a previous round of review and you feel that this manuscript is now acceptable for publication, you may indicate that here to bypass the “Comments to the Author” section, enter your conflict of interest statement in the “Confidential to Editor” section, and submit your "Accept" recommendation.

Reviewer #1: (No Response)

Reviewer #2: All comments have been addressed

2. Is the manuscript technically sound, and do the data support the conclusions?

Reviewer #1: Yes

Reviewer #2: Yes

3. Has the statistical analysis been performed appropriately and rigorously? 

Reviewer #1: Yes

Reviewer #2: Yes

4. Have the authors made all data underlying the findings in their manuscript fully available?

Reviewer #1: Yes

Reviewer #2: Yes

5. Is the manuscript presented in an intelligible fashion and written in standard English?

Reviewer #1: Yes

Reviewer #2: Yes

6. Review Comments to the Author

Reviewer #1: Comparing Figure 6 with Figure 7 and Fig.1 , I think the positions of thermocouples and microwave antennas are fixed in every experiments, so there is no ” However, due to the different distances of the seven measurement points from the high-temperature center“ as author explained. On the contrary, if the positions of the two are not fixed, is the temperature measurement point data in machine learning reliable?

Reviewer #2: (No Response)

7. PLOS authors have the option to publish the peer review history of their article (what does this mean?). If published, this will include your full peer review and any attached files.

Reviewer #1: No

Reviewer #2: No

---

## [Editor Report · Acceptance letter]

16 Sep 2024

PONE-D-24-15770R1 

PLOS ONE

Dear Dr. Huang, 

I'm pleased to inform you that your manuscript has been deemed suitable for publication in PLOS ONE. Congratulations! Your manuscript is now being handed over to our production team.

Kind regards, 

on behalf of

Dr. Carlos Alberto Antunes Viegas 

Academic Editor

PLOS ONE